

# Evolutionary radiation of earless frogs in the Andes: molecular phylogenetics and habitat shifts in high-elevation terrestrial breeding frogs

Rudolf von May[1,2], Edgar Lehr[3] and Daniel L. Rabosky[1]

[1] Museum of Zoology & Department of Ecology and Evolutionary Biology, University of Michigan—Ann Arbor, Ann Arbor, MI, United States of America
[2] Museum of Vertebrate Zoology, University of California, Berkeley, Berkeley, CA, United States of America
[3] Department of Biology, Illinois Wesleyan University, Bloomington, IL, United States of America

Corresponding author
Rudolf von May,
rvonmay@gmail.com,
rvonmay@umich.edu

## ABSTRACT

The loss of hearing structures and loss of advertisement calls in many terrestrial breeding frogs (Strabomantidae) living at high elevations in South America are common and intriguing phenomena. The Andean frog genus *Phrynopus* Peters, 1873 has undergone an evolutionary radiation in which most species lack the tympanic membrane and tympanic annulus, yet the phylogenetic relationships among species in this group remain largely unknown. Here, we present an expanded molecular phylogeny of *Phrynopus* that includes 24 nominal species. Our phylogeny includes *Phrynopus peruanus*, the type species of the genus, and 10 other species for which genetic data were previously unavailable. We found strong support for monophyly of *Phrynopus*, and that two nominal species—*Phrynopus curator* and *Phrynopus nicoleae*—are junior synonyms of *Phrynopus tribulosus*. Using X-ray computed tomography (CT) imaging, we demonstrate that the absence of external hearing structures is associated with complete loss of the auditory skeletal elements (columella) in at least one member of the genus. We mapped the tympanum condition on to a species tree to infer whether the loss of hearing structures took place once or multiple times. We also assessed whether tympanum condition, body size, and body shape are associated with the elevational distribution and habitat use. We identified a single evolutionary transition that involved the loss of both the tympanic membrane and tympanic annulus, which in turn is correlated with the absence of advertisement calls. We also identified several species pairs where one species inhabits the Andean grassland and the other montane forest. When accounting for phylogenetic relatedness among species, we detected a significant pattern of increasing body size with increasing elevation. Additionally, species at higher elevations tend to develop shorter limbs, shorter head, and shorter snout than species living at lower elevations. Our findings strongly suggest a link between ecological divergence and morphological diversity of terrestrial breeding frogs living in montane gradients.

## INTRODUCTION

The loss of hearing structures has occurred multiple times throughout the evolutionary history of anurans (*Jaslow, Hetherington & Lombard, 1988*; *Boistel et al., 2013*; *Pereyra et al., 2016*). Multiple losses and regains of the tympanic middle ear in anurans involve the outermost elements of the middle ear, including the tympanic membrane (a thin disk composed of non-glandular skin), the tympanic annulus (a ring composed of cartilage), and, less frequently, the columella (a bone also known as stapes; *Vorobyeva & Smirnov, 1987*; *Pereyra et al., 2016*). Within the middle ear cavity, the outermost portion of the columella is in contact with the inner wall of the tympanic membrane and aids in the transmission of airborne vibrational signals to the inner ear (*Mason et al., 2015*). Earlessness (i.e., the loss of tympanic middle ear structures) is particularly common in some groups such as the true toad family Bufonidae, which contains over 200 earless species (*Pereyra et al., 2016*) or approximately 30% of the total species richness of this group (607 species; *AmphibiaWeb, 2017*). In South America, the absence of tympanic membrane and tympanic annulus is also common in the terrestrial breeding frog family Strabomantidae (*Hedges, Duellman & Heinicke, 2008*; *Padial, Grant & Frost, 2014*) and is particularly prevalent in several clades distributed at high elevations (e.g., *Bryophryne*, *Phrynopus*; *Duellman & Lehr, 2009*). How these frogs communicate and what factors drove the loss of hearing structures at high elevations are questions that remain unexplored in this diverse group of amphibians.

Given that the tympanic middle ear has been lost multiple times in the evolutionary history of anurans, including complex patterns of loss and regain in some large radiations (e.g., bufonids; *Pereyra et al., 2016*), it is relevant to examine how the loss of tympanic middle ear structures is distributed across other less known but equally diverse radiations such as Strabomantidae. Understanding the potential causes and ecological consequences of repeated losses and occasional regains of a character across a phylogenetic tree could be useful to infer patterns of morphological and ecological diversity in this group. Within this family, the Andean frog genus *Phrynopus* (Fig. 1) represents a special case where the large majority of species (>90%) have lost the tympanic membrane and tympanic annulus. Members of this clade inhabit montane forests and Andean grasslands (also known as Puna) at elevations between 2,600 and 4,490 m (*Duellman & Lehr, 2009*; *Lehr & von May, 2017*; *Rodríguez & Catenazzi, 2017*), and share morphological features with other high elevation strabomantids in the genera *Bryophryne*, *Pristimantis*, *Lynchius*, and *Oreobates* (*Lehr & von May, 2017*). Given that the Andes have experienced multiple uplift events since the Miocece (*Gregory-Wodzicki, 2000*), the resulting combined effects of habitat, elevation, and climatic gradients played an important role in species' ecological divergence along montane gradients (*Moritz et al., 2000*; *Rheindt, Christidis & Norman, 2008*; *Antonelli et al., 2009*; *Irestedt et al., 2009*; *von May et al., 2017*) and may have also influenced the morphological diversity of terrestrial breeding frogs observed today.

The genus *Phrynopus* currently contains 34 species distributed in the Peruvian Andes (*Lehr & Rodriguez, 2017*; *Rodríguez & Catenazzi, 2017*). From these, only three species exhibit a tympanic membrane and tympanic annulus whereas the remaining 31 species lack both structures (we refer to these as earless species, i.e., those with no tympanic

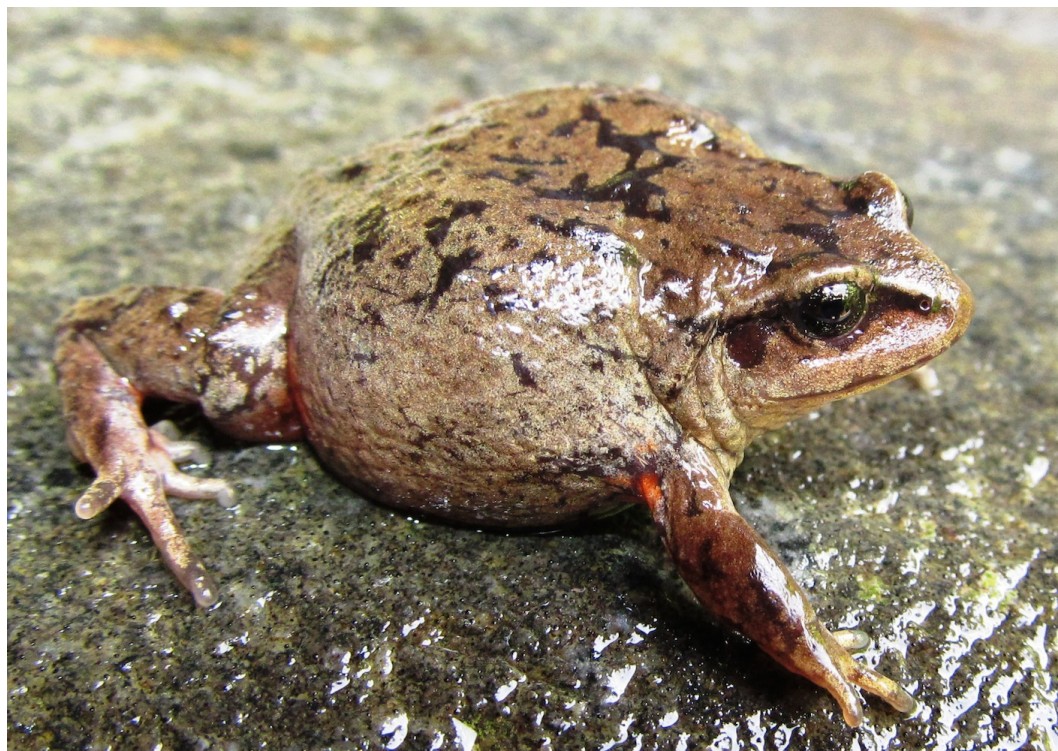

**Figure 1** **Adult female of *Phrynopus peruanus* MUSM 38315 collected at the type locality in central Peru.** Notice the presence of tympanum and the relatively short limbs and short head compared with other species of *Phrynopus*. Photograph by R von May.

membrane and no tympanic annulus). Additionally, only the eared species are known to produce advertisement calls (i.e., earless species do not produce calls; *Duellman & Lehr, 2009*; E Lehr & R von May, pers. obs., 2012–2014). In contrast, in other anuran families that contain lineages that lost the tympanic middle ear, most of those lineages typically retained the ability to call and species evolved alternative extra-tympanic sensory pathways that enable the transmission of low frequency sound to the inner ear (e.g., Bombinatoridae, *Hetherington & Lindquist, 1999*; Brachycephalidae, *Goutte et al., 2017*; Bufonidae, *Pereyra et al., 2016*; *Womack et al., 2017*; Sooglossidae, *Boistel et al., 2013*). While the actual mode of communication in the 31 earless species of *Phrynopus* is currently unknown, it is valuable to examine the presence and absence of tympanic middle ear structures in a phylogenetic framework to understand if the loss of hearing structures took place once or multiple times in this clade. Speciation in the absence of common acoustic signals in frogs, such as the one observed in *Phrynopus*, raises questions about the factors that affect speciation given that advertisement calls are recognized as a premating isolation mechanism (*Boughman, 2002*).

Here, we present an expanded molecular phylogeny of *Phrynopus* that contains 53 terminals representing 24 species. We inferred a species tree, and we used this tree to examine the relationship between species' morphological features and their elevational distributions in a phylogenetic comparative framework. To determine whether key skeletal

traits (columella) were lost, in addition to other tympanic structures that are observable with a dissecting scope, we used X-ray micro-computed tomography (μCT) to obtain images for one eared and one earless species. *Phrynopus* comprise a radiation lacking both the communication signal (advertisement calls) and sensory system (tympanic membrane and annulus) typical to the large majority of anurans. Previous studies focusing on other anuran taxa have shown that earless species live in diverse habitats and may experience different selection pressures, suggesting that the loss of tympanic middle ear is not associated with a particular environment (*Vorobyeva & Smirnov, 1987*; *Smirnov, 1991*; *Womack et al., 2017*). Overall patterns of habitat use and elevational distribution of *Phrynopus* also seem to support this idea, but proper testing is required.

Given that terrestrial breeding frogs living at high elevations experience different ecological conditions than those living at lower elevations (*Catenazzi, Lehr & Vredenburg, 2014*; *von May et al., 2017*), they may also experience different selection pressures promoting morphological diversification. To test this hypothesis, we examined if body size and body shape vary across habitats or elevations. Several studies have suggested that larger body size in Neotropical terrestrial breeding frogs might represent an adaptation to high-elevation habitats (*Hedges, 1999*; *Gonzalez-Voyer et al., 2011*). Given that species living in high-elevation habitats such as the Andean grassland appear to have larger body and different body shape than species living at lower elevations (*Lehr & von May, 2017*), we hypothesized that body size and body shape are associated with the type of habitat use and elevation. These direct-developing frogs are amenable for testing this hypothesis because they have low vagility (resulting in local genetic structuring) and in most cases limited geographic and elevational ranges (*Duellman & Lehr, 2009*; *De la Riva et al., 2017*; *von May et al., 2017*). A larger body size at higher elevations would agree with Bergmann's rule, which predicts that body size increases in cooler environments (*Bergmann, 1847*; *Mayr, 1956*); this pattern requires further testing because it is not universal among amphibians (*Adams & Church, 2007*). We did not expect to observe different morphologies between eared and earless species, because the absence of tympanic middle ear in *Phrynopus* is not associated with a particular environment (*Duellman & Lehr, 2009*).

The first molecular phylogeny of *Phrynopus* (*Lehr, Fritzsch & Müller, 2005*) included eight currently recognized species in addition to five other species that were subsequently transferred to different genera (e.g., *Phrynopus iatamasi* was transferred to the genus *Psychrophrynella* by *Hedges, Duellman & Heinicke, 2008*, and it was subsequently transferred to the genus *Microkayla* by *De la Riva et al., 2017*). *Hedges, Duellman & Heinicke (2008)* inferred that *Phrynopus* is a close relative of the clades composed of *Lynchius* and *Oreobates*, a result corroborated by *Padial, Grant & Frost (2014)*. Both *Hedges, Duellman & Heinicke (2008)* and *Padial, Grant & Frost (2014)* included sequences of 11 species of *Phrynopus* in their analyses. More recently, a molecular phylogenetic analysis focusing on Holoadeninae (the strabomantid subfamily that contains *Phrynopus*) included sequences of 13 species of *Phrynopus* (*De la Riva et al., 2017*). *De la Riva et al. (2017)* reported high genetic similarity between two species, *P. nicoleae* and *P. tribulosus*, and suggested a possible synonymy, but no formal taxonomic action was proposed. Furthermore, none of the previous analyses included *Phrynopus peruanus* (*Peters, 1873*), the type species of

the genus, despite the rediscovery and collection of specimens at the type locality (*Lehr, 2007a*). Given that multiple taxonomic changes have taken place in this group (e.g., some *Phrynopus* transferred to *Psychrophrynella* and subsequently to *Microkayla*; *Phrynopus ayacucho* transferred to the genus *Oreobates*; *Padial, Grant & Frost, 2014*), it is essential to include the type species in successive phylogenetic analyses such as those presented here.

## MATERIALS AND METHODS

### Study area and field surveys

We surveyed multiple montane sites between April 2012 and September 2014, and collected specimens and ecological data on 11 species of *Phrynopus*. The surveyed areas included the type locality of *Phrynopus badius* Lehr, Moravec & Cusi, 2012, *Phrynopus curator* Lehr, Moravec & Cusi, 2012, *Phrynopus daemon* Chávez et al., 2015, *Phrynopus interstinctus* Lehr & Oróz, 2012, *Phrynopus inti* Lehr et al., 2017, *Phrynopus juninensis* (Shreve, 1938), *Phrynopus montium* (Shreve, 1938), *Phrynopus peruanus* Peters, 1873, *Phrynopus unchog* Lehr & Rodríguez, 2015, *Phrynopus vestigiatus* Lehr & Oróz, 2012, *Phrynopus* spI (R von May, 2018, unpublished data). In addition to collecting primary data on these species, we reviewed published studies to assemble a dataset containing minimum and maximum elevation, elevational midpoint, and habitat (Table S1) for those species included in phylogenetic comparative analyses (see below). Research and collecting permits were issued by the Dirección General Forestal y de Fauna Silvestre (DGFFS) and the Servicio Nacional Forestal y de Fauna Silvestre (120-2012-AG-DGFFS-DGEFFS, 064-2013-AG-DGFFS-DGEFFS, 292-2014-AG-DGFFS-DGEFFS, R.D.G. No 029-2016-SERFOR-DGGSPFFS, R.D.G. 405-2016-SERFOR-DGGSPFFS, and Contrato de Acceso Marco a Recursos Genéticos, No 359-2013-MINAGRI-DGFFS-DGEFFS) and the Servicio Nacional de Areas Naturales Protegidas (No 001-2012-SERNANP-JEF). Export permits were issued by the Ministerio del Ambiente, Lima, Peru. Use of vertebrate animals was approved by the Animal Care and Use Committee of the University of California (ACUC #R278-0412, R278-0413, and R278-0314). Specimens collected during this study were deposited in the herpetological collection at the Museo de Historia Natural Universidad Nacional Mayor de San Marcos (MUSM), Lima, Peru.

### Morphological data

We compiled available morphological data for 21 species of *Phrynopus* included in our phylogenetic analysis (data were not available for *Phrynopus* sp, an undescribed species from Pasco region, central Peru; *Lehr et al., 2017*). Most of these data were extracted from the original publications describing these species (*Hedges, 1990*; *Lehr, Köhler & Ponce, 2000*; *Lehr, 2001*; *Lehr & Aguilar, 2002*; *Lehr & Aguilar, 2003*; *Lehr, Lundberg & Aguilar, 2005*; *Lehr, Fritzsch & Müller, 2005*; *Chaparro, Padial & De la Riva, 2008*; *Duellman & Hedges, 2008*; *Lehr & Oróz, 2012*; *Lehr, Moravec & Cusi, 2012*; *Chávez et al., 2015*; *Lehr & Rodriguez, 2017*; *Lehr & von May, 2017*; *Rodríguez & Catenazzi, 2017*). Additionally, for three species (*P. juninensis*, *P. peruanus,* and *Phrynopus* spI), we took measurements of specimens collected in recent expeditions. Our dataset included only adult individuals ($N = 76$); for all of these, sex and maturity of specimens were determined by observing gonads through

dissections under a stereomicroscope (Table S1; see original species descriptions for further detail). Our dataset included the following measurements: snout–vent length (SVL), tibia length (TL, distance from the knee to the distal end of the tibia), foot length (FL, distance from proximal margin of inner metatarsal tubercle to tip of Toe IV), head length (HL, from angle of jaw to tip of snout), head width (HW, at level of angle of jaw), horizontal eye diameter (ED), interorbital distance (IOD), upper eyelid width (EW), internarial distance (IND), eye–nostril distance (E-N, straight line distance between anterior corner of orbit and posterior margin of external nares). Information on species tympanum condition was obtained from *Duellman & Lehr (2009)* and from subsequent species descriptions (*Lehr & Oróz, 2012*; *Lehr, Moravec & Cusi, 2012*; *Chávez et al., 2015*; *Lehr & Rodriguez, 2017*; *Lehr et al., 2017*; *Rodríguez & Catenazzi, 2017*).

## Micro-computed tomography

To determine the condition of the columella, a key element of the tympanic middle ear, we obtained X-ray μCT images for one eared and one earless species, *Phrynopus peruanus* and *Phrynopus montium*, respectively. Two voucher specimens stored in ethanol were scanned in the Micro-CT Core facility at the University of Michigan. Specimens were placed in a 34 mm diameter specimen holder and scanned using a microCT system (μCT100; Scanco Medical, Bassersdorf, Switzerland). Scan settings were: voxel size 11.4 μm, 55 kVp, 145 μA, 0.5 mm AL filter, 1,000 projections around 180°, integration time of 1,000 ms and 3 average data. Data were exported to DICOM files using Scanco's proprietary software. We obtained images after segmentation using isosurface representations and three-dimensional renderings using the Amira-Avizo software. According to the predictions of *Pereyra et al. (2016)* the presence of tympanic membrane and tympanic annulus in *P. peruanus* implies that the columella is present. Thus, we expected to observe the columella in a CT scan of *P. peruanus*. In contrast, the absence of both tympanic membrane and tympanic annulus in *P. montium* did not imply that the columella would be absent. Thus, CT scanning this specimen was particularly critical to determine if the columella was present (or not) in this species.

## Molecular phylogenetic analysis

Our analysis included DNA sequence data from *Phrynopus* species that were available in GenBank (as of 1 August 2017; Table S2) as well as sequences from other closely related genera (*Lynchius*, *Oreobates*) and more distantly related ones (*Ischnocnema guentheri*, *Hypodactylus brunneus*, and *Hypodactylus dolops*) as outgroups following the results of *Padial, Grant & Frost (2014)*. Additionally, newly produced sequences include those obtained from 11 species: *P. badius*, *P. curator*, *P. daemon*, *P. interstinctus*, *P. juninensis*, *P. montium*, *P. peruanus*, *P. unchog*, *P. vestigiatus*, *Phrynopus inti*, and *Phrynopus* spI. A notable addition to this phylogeny is *P. peruanus*, the type species of the genus.

The mitochondrial genes were a section of the 16S rRNA gene, a section of the 12S rRNA gene, and the protein-coding gene cytochrome c oxidase subunit I (COI). The nuclear genes were the recombination-activating protein 1 (RAG1) and Tyrosinase precursor (Tyr). Extraction, amplification, and sequencing of DNA followed protocols

previously used for Neotropical terrestrial breeding frogs (*Lehr, Fritzsch & Müller, 2005*; *Hedges, Duellman & Heinicke, 2008*). Primers used are listed in Table S3. We employed the following thermocycling conditions to amplify DNA from each gene using the polymerase chain reaction (PCR). For 16S, we used: 1 cycle of 96 °C/3 min; 35 cycles of 95 °C/30 s, 55 °C/45 s, 72 °C/1.5 min; 1 cycle 72 °C/7 min. For 12S, we used: 1 cycle of 94 °C/1.5 min; 35 cycles of 94 °C/45 s, 50 °C/1 min., 74 °C/2 min; 1 cycle 72 °C/10 min. For COI, we used: 1 cycle of 96 °C/3 min; 35 cycles of 95 °C/30 s, 55 °C/45 s, 72 °C/1.5 min; 1 cycle 72 °C/7 min. For RAG1, we used: 1 cycle of 96 °C/2 min; 40 cycles of 94 °C/30 s, 52 °C/30 s, 72 °C/1.5 min; 1 cycle 72 °C/7 min. For Tyr, we used: 1 cycle of 94 °C/5 min; 40 cycles of 94 °C/30 s, 54 °C/30 s, 72 °C/1 min; 1 cycle 72 °C/7 min. We completed the cycle sequencing reactions by using the corresponding PCR primers and the BigDye Terminator 3.1 (Applied Biosystems, Foster City, CA, USA), and obtained sequence data by running the purified reaction products in an ABI 3730 Sequence Analyzer (Applied Biosystems, Foster City, CA, USA). We deposited the newly obtained sequences in GenBank (Table S2).

We used Geneious R6, version 6.1.8 (*Biomatters, 2013*; http://www.geneious.com/) to align the sequences with the built-in multiple alignment program. Prior to conducting phylogenetic analysis, we used PartitionFinder, version 1.1.1 (*Lanfear et al. 2012*) to select the appropriate models of nucleotide evolution and used the Bayesian information criterion (BIC) to determine the best partitioning scheme and substitution model for each gene. We divided the partitions a priori; each mitochondrial gene was considered a data block whereas each nuclear gene was divided by codon position (i.e., three data blocks per gene). The best partitioning schemes and models of evolution are included in the Results section.

We employed a Bayesian approach using MrBayes, version 3.2.0 (*Ronquist & Huelsenbeck, 2003*) to infer a molecular phylogeny. Our analysis included 72 terminals and a 2,646-bp concatenated partitioned dataset. We performed an MCMC Bayesian analysis that consisted of two simultaneous runs of eight million generations, and we set the sampling rate to be once every 1,000 generations. Each run had three heated chains and one "cold" chain, and the burn-in was set to discard the first 25% samples from the cold chain. Following the completion of the analysis, we used Tracer 1.6 (*Rambaut & Drummond, 2007*) to verify convergence. Subsequently, we used FigTree (http://tree.bio.ed.ac.uk/software/figtree/) to visualize the majority-rule consensus tree and the posterior probability values to assess node support.

We also used a multispecies coalescent approach implemented in *BEAST v1.6.2 (*Drummond & Rambaut, 2007*) to infer a species tree to be used for phylogenetic comparative analyses (see below). Our analyses only depend on the relative branch lengths of the tree, but we preferred to illustrate our tree in approximate units of time. Therefore, we used an uncorrelated relaxed molecular clock with the rate of nucleotide substitution for 16S set at 1% per million years as done in a recent study (*von May et al., 2017*). However, as in *von May et al. (2017)*, we note that the dates associated with the tree should only be viewed as very approximate and that there can be multiple sources of error when calibrating phylogenies (*Arbogast et al., 2002*). The analysis in *BEAST included two independent runs, each with 100 million generations and sampled every 10,000 generations. Following the completion of the analysis, we used Tracer v1.5 (*Rambaut & Drummond,*

*2007*) to examine effective sample sizes, verify convergence of the runs, and to ensure the runs had reached stationarity. Observed effective sample sizes were sufficient for most parameters (ESS > 200) except for substitution rates for a few partitions. We discarded the first 10% of samples from each run as burn-in. Subsequently, we used LogCombiner v1.6.2 to merge all remaining trees from both runs and used TreeAnnotator v1.6.2 (*Drummond & Rambaut, 2007*) to summarize trees and obtain a Maximum Clade Credibility tree. We visualized the Maximum Clade Credibility (MCC) tree and the associated node support values in FigTree (http://tree.bio.ed.ac.uk/software/figtree/).

## Morphological and ecological divergence

We mapped the tympanum condition on to the species tree to infer whether the loss of hearing structures took place once or multiple times. The hypothesized ancestral tympanic condition in *Phrynopus* along the internal branches of a phylogenetic tree including members of two other closely related genera (*Lynchius* and *Oreobates*) was determined using Bayesian stochastic character mapping (*Huelsenbeck, Nielsen & Bollback, 2003*) implemented in the phytools package (*Revell, 2012*). A mapping of tympanum presence/absence in the phylogeny was run on 500 stochastically mapped trees to take into account the uncertainty in the ancestral tympanum condition.

We also mapped the elevational range on to the species tree to visually assess the patterns of elevational distribution, phylogenetic relatedness, and presence/absence of tympanic membrane and annulus. For visualization purposes, we modified the range of species known from a single elevation or those known from a very narrow range (<50 m) to cover a minimum elevational band of 80 m. For species known from a single elevation, this elevation was assumed to be the midpoint and this number was used to expand the range on both directions (−40 and +40) to be 80 m. This allowed us to plot a color-coded bar representing the main habitat of each species. For species known to cover a narrow elevational band (<50 m), we first calculated the elevational midpoint (i.e., maximum elevation—minimum elevation) and used this value to expand the range on both directions (−40 and +40) to be 80 m. These adjusted ranges were used in only one figure (i.e., original data were used in all other figures and analyses).

Additionally, we examined if body size and body shape vary across habitats or elevations. Given that terrestrial breeding frogs living in high-elevation habitats appear to have larger body size and different body shape than species living at lower elevations (*Hedges, 1999*; *Gonzalez-Voyer et al., 2011*; *Lehr & von May, 2017*), we hypothesized that body size and body shape in *Phrynopus* are associated with the type of habitat use and elevation. First, we used phylogenetic principal components analysis (PCA) to examine morphological diversity between eared and earless species and among habitats. To remove the possible confounding effect of body size difference among species while controlling for non-independence resulting from shared phylogenetic history, we performed a phylogenetic size-correction using generalized least-squares (GLS) regression analysis (*Revell, 2009*). In this analysis, we considered SVL as the independent variable and all other morphological data were considered dependent variables. To perform the phylogenetic PCA, we pruned the species tree to only include 21 species of *Phrynopus* with corrected morphological
data. We projected the first two PC axes on a phylomorphospace using the function phylomorphospace in phytools (*Revell, 2012*). We also used a phylogenetic ANOVA (*Garland Jr et al., 1993*) to quantify shape differences between eared and earless species and to quantify shape differences among habitats. For the latter comparison, we defined three groups (Andean grassland, montane forest, both habitats) and selected a posthoc test to compare means among groups; we used the "holm" method to adjust the P-values to account for multiple comparisons. These analyses were also implemented using phytools.

We used a scatterplot pairwise matrix to examine the relationships among body size, other body size-corrected morphometric variables, and elevation. We used the R package phylolm (*Ho & Anné, 2014a*; *Ho & Anné, 2014b*) to fit phylogenetic generalized linear regression models (PGLS), implemented under three models of evolution: (a) pure ordinary least squares (OLS) regression (i.e., no phylogenetic signal), (b) pure Brownian motion (BM) model, and (c) lambda transform model, which allows intermediate levels of phylogenetic signal between pure BM and pure OLS (we reported Pagel's lambda coefficient, $\lambda$, for the third model). We used the AIC value to identify the model that best explains the variation in observed data (*Ho & Anné, 2014b*).

# RESULTS

## Micro-computed tomography

We visualized three-dimensional reconstructions of X-ray μCT images of the skull of the eared taxon *Phrynopus peruanus*, and confirmed the presence of the columella in this species (Fig. 2). In contrast, the columella has been completely lost in the earless *P. montium* (Fig. 2). This observation, in combination with previous findings (*Trueb & Lehr, 2008*), confirms that *P. montium* has lost the three main elements of the middle ear (tympanic membrane, tympanic annulus, and columella).

## Molecular phylogenetic analysis

The best partitioning scheme included six subsets (BIC value: 34189.93). The first partition subset included both the 12S and 16S sequences and the best fitting substitution model was GTR + I + G. The remaining five subsets were partitioned according to codon positions as follows (substitution model in parenthesis): one set including the 1st codon position of COI (K80 + I), one set including the 2nd codon position of COI (HKY + I), one set including the 3rd codon position of COI (GTR + G), one set including the 1st and 2nd codon position of RAG1 and the 1st and 2nd codon position of Tyr (HKY + I), and one set including the 3rd codon position of both RAG1 and Tyr (HKY + G). At the end of the MCMC run, the average standard deviation of split frequencies was 0.002538. We found that *Phrynopus* is a well-supported monophyletic group that is closely related to *Oreobates* and *Lynchius* (Fig. 3). The earliest divergence within *Phrynopus* occurred between a clade formed by two species, *P. peruanus* and an undescribed species (*Phrynopus* spI), and all other species. Our analysis suggested that two nominal species, *Phrynopus curator* and *Phrynopus nicoleae*, are junior synonyms of *Phrynopus tribulosus*; a proposed formal taxonomic change is included below (New synonymy).

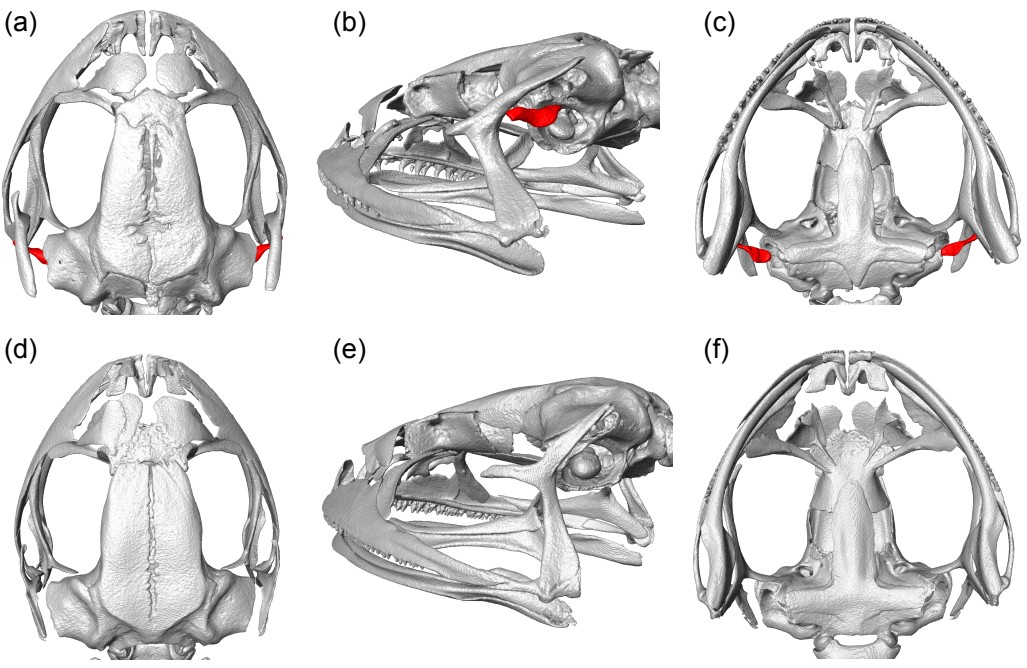

**Figure 2** **CT scan images.** Three-dimensional reconstructions based on μCT data, from the skull and middle ear structures in eared and earless frogs. Dorsal (A), postero-lateral (B) and ventral (C) views of the skull of *Phrynopus peruanus* MUSM 38315, and the presence of the columella (stapes) indicated in red. Dorsal (D), postero-lateral (E) and ventral (F) views of the skull of *Phrynopus montium* MUSM 33259; the columella is absent in this species.

## Phylogenetic relatedness and trait-and-habitat data

We used the species tree obtained with *BEAST to visually assess tympanum presence/absence (Fig. 4). Our analysis identified a single evolutionary transition that involved the loss of both the tympanic membrane and tympanic annulus (Fig. 4).

We used the species tree to assess the patterns of elevational distribution and phylogenetic relatedness (Fig. 5). We observed substantial differences in elevational distribution among closely-related species, with some species pairs occurring in distinct habitats (Fig. 5). In four cases, close relatives had non-overlapping elevational distributions (*P. peruanus–Phrynopus* spI; *P. bracki–P. badius*; *P. miroslawae–P. tautzorum*; *P. unchog–P. daemon*). Additionally, in three cases, close relatives were associated with either Andean grassland or montane forest (*P. peruanus—Phrynopus* spI; *P. barthlenae–P. horstpauli*; *P. miroslawae–P. tautzorum*).

Our analysis of body-size corrected morphological data showed that *Phrynopus* species exhibit a considerable variation in body shape. This variation appeared not to be associated with the presence or absence of tympanic middle ear structures (Figs. 6A and 6B); instead, species differences in body shape were associated with the habitat they occupy (Figs. 6C and 6D). Specifically, species that live in Andean grassland occupied a smaller area in the phylomorphospace projection that species that live in montane forest or those that occupy the two habitats.
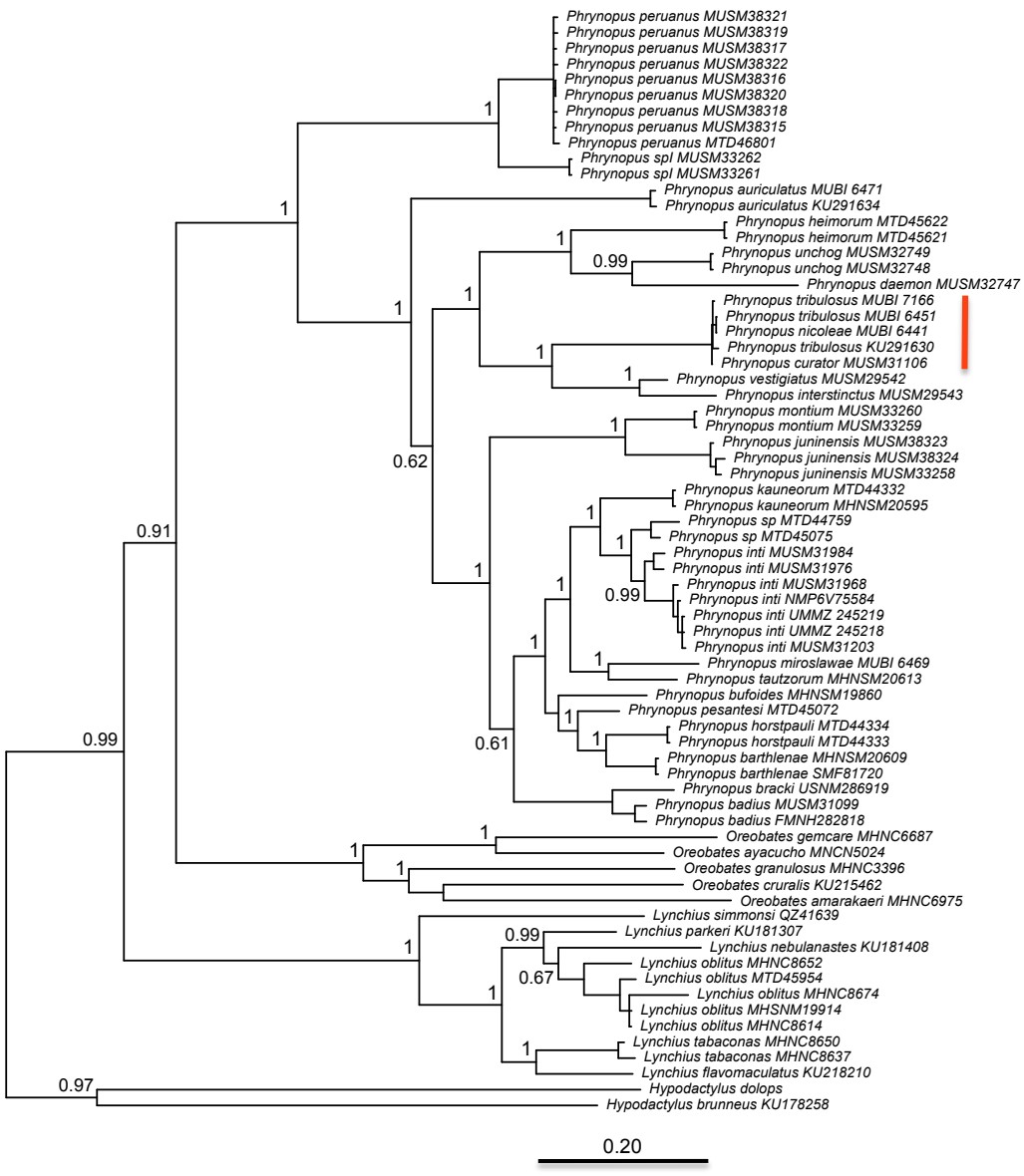

**Figure 3 Phylogeny.** Bayesian maximum clade-credibility tree for species included in this study based on a 2,646-bp concatenated partitioned dataset (16S, 12S, COI, RAG1, Tyr) analyzed in MrBayes (posterior probabilities are indicated at each node). The red bar indicates terminals representing *Phrynopus tribulosus*; two of these were recognized as *Phrynopus nicoleae* and *Phrynopus curator* in previous studies (see text for details).

Considering data from all adult specimens (males and females), the first seven principal components were found to explain 97.5% of the variance within the size-corrected morphological dataset following a phylogenetic size-correction using GLS regression (Table 1). Considering data from female specimens only, the first seven principal components were found to explain 96.7% of the variance within the size-corrected morphological dataset following a phylogenetic size-correction using GLS regression

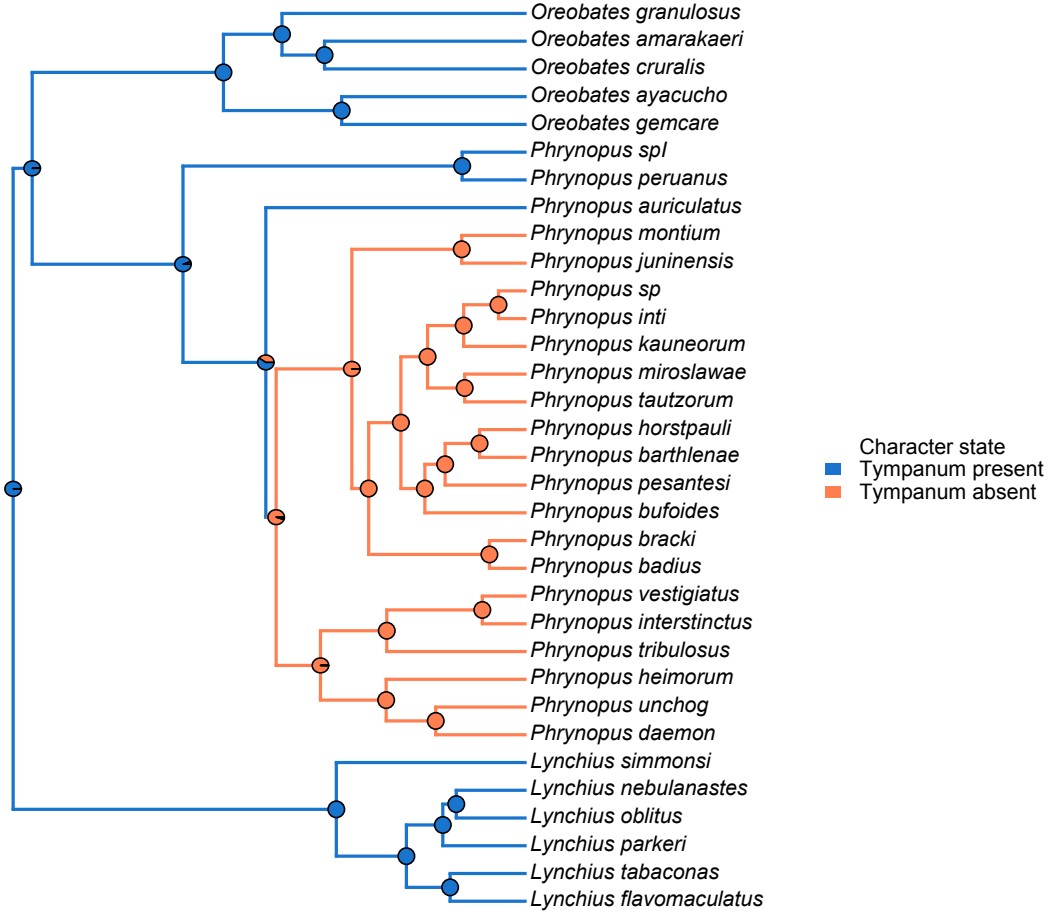

**Figure 4** **Stochastic mapping tree.** Species tree of *Phrynopus* species included in this study, and representative species of *Lynchius* and *Oreobates*, with a sample stochastic character map of tympanum condition (blue indicates presence and orange indicates absence of tympanic annulus) and ancestral state estimates from 500 simmaps indicated at the nodes.

(Table 2). We found no shape differences between eared and earless species (phylogenetic ANOVA, all tests $P > 0.10$). We found that habitat had a significant effect on shape, notably in those variables (FL, HL, HW, IOD, E-N) that had greater contribution on PC2 (phylogenetic ANOVA, $F = 3.943$, $P = 0.036$).

We used a pairwise scatterplot matrix to visualize the cross-correlations among body-size corrected morphological data, SVL, and elevation (Fig. S1). Our PGLS analyses showed that there is a significant pattern of increasing body size at higher elevations, and that species living at higher elevations tend to develop shorter limbs, shorter head, and shorter snout than those species living at lower elevations (Table 3, Fig. 7). All tested relationships were significant both when controlling for phylogeny under the lambda transform model and also under pure OLS regression, and OLS was the best-supported model in five cases. One relationship, total length vs. elevational midpoint, was best supported under the Brownian motion (BM) model (and it was also significant under both the OLS and Lambda models).

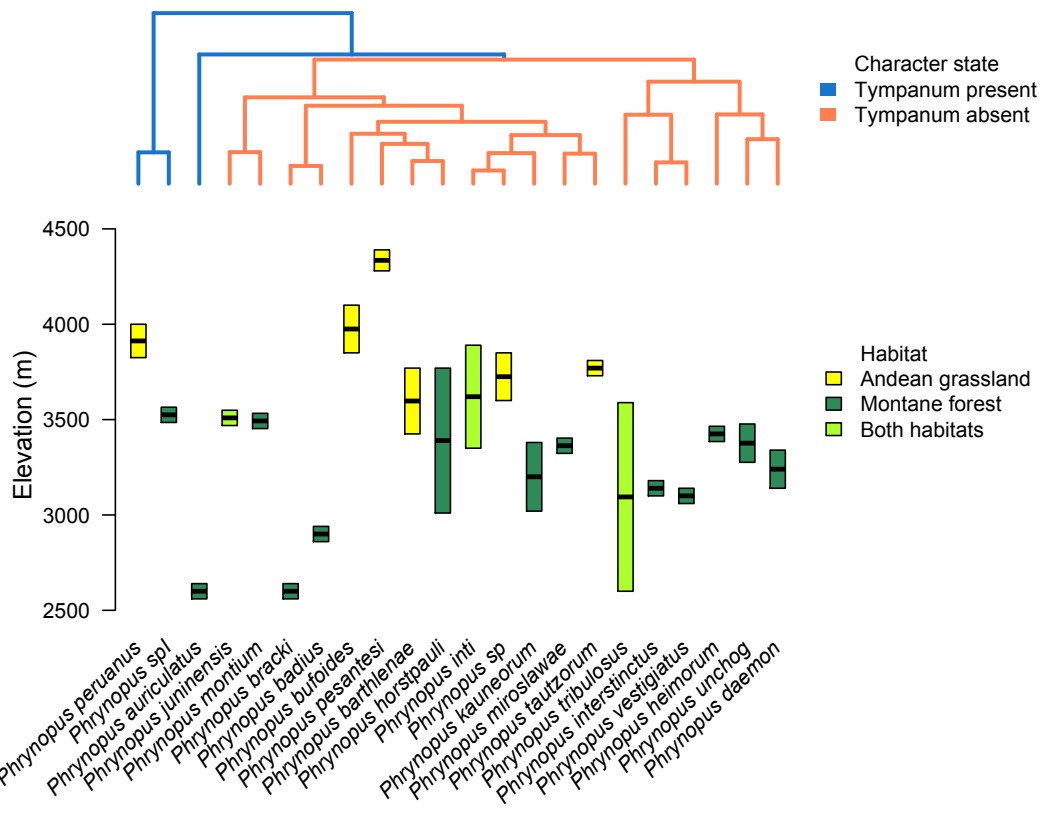

**Figure 5** **Tree and elevational ranges.** Elevational distribution and habitat use are not related to tympanum condition in *Phrynopus*. The species tree (obtained with *BEAST) depicting the relationship among 22 species of *Phrynopus* (top) is color-coded according to the presence (blue) or absence (orange) of tympanic membrane and annulus. The elevational ranges (bottom) are color-coded according to species' primary habitat: Andean grassland (yellow), montane forest (dark green), and both habitats (light green). The elevational midpoint is denoted by a black bar.

## New synonymy

Here we synonymize both *Phrynopus curator Lehr, Moravec & Cusi, 2012* and *Phrynopus nicoleae Chaparro, Padial & De la Riva, 2008* with *Phrynopus tribulosus Duellman & Hedges, 2008*.

### *Phrynopus tribulosus Duellman & Hedges, 2008*

*Phrynopus nicoleae, Chaparro, Padial & De la Riva, 2008*; p. 53, Figs. 5 and 6 (MHNC 6441(holotype) collected by JC Chaparro, A Quiroz and D Salcedo at Santa Bárbara (10°20′36.3″S, 75°38′17.9″W; 3,589 m a.s.l.), Distrito de Huancabamba, Provincia de Oxapampa, Departmento Pasco, Peru, 2007)—*Lehr & Oróz, 2012*, p. 55, 59; *Lehr, Moravec & Cusi, 2012*, p. 51, 52, 55, 59, 61, 65, 66, Table 1; *Padial, Grant & Frost, 2014*, p. 124; *Chávez et al., 2015*, p. 21; *De la Riva et al., 2017*, p. 5, 31, 44; *Lehr & Rodriguez, 2017*, p. 333, 334; *Rodríguez & Catenazzi, 2017*, p. 383, 386, 390, 399, 401, 403, Table 2.

*Phrynopus curator Lehr, Moravec & Cusi, 2012*; p. 63, Figs. 5 and 7, 8, 10 (MUSM 31106 (holotype) collected by E Lehr, J Moravec, and JC Cusi at Quebrada Yanachaga

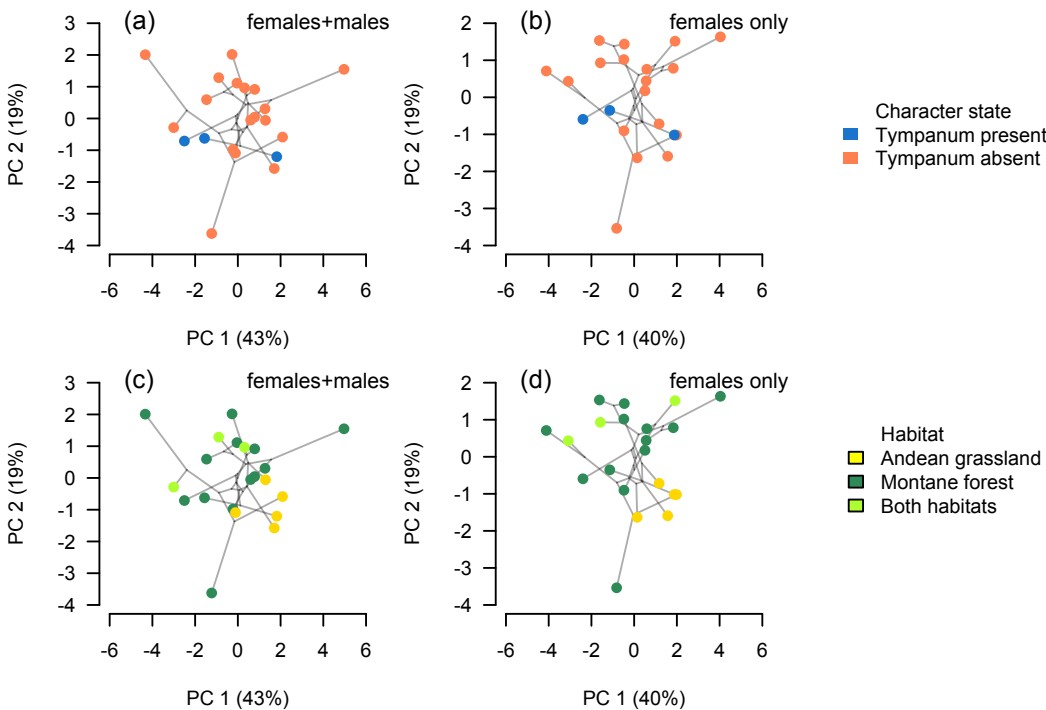

**Figure 6** **Phylomorphospace.** Phylomorphospace projection of *Phrynopus* species based on body-size corrected data following a phylogenetic size-correction using generalized least-squares (GLS) regression analysis. Body shape in *Phrynopus* is not associated with tympanum condition (A, B). In contrast, species' body shape appears to be related to habitat: species that inhabit the Andean grassland occupy a smaller region of morphospace than species that live in montane forest (C, D). The full dataset included data from adult males and females (A, C); the reduced dataset included adult females only (B, D).

**Table 1** **PCA loadings full dataset.** Loadings for the nine morphological variables from a phylogenetic PCA ran on body-size corrected residuals following a phylogenetic size-correction using GLS regression. The first seven PCs explained 97.5% of the variance, as estimated using the full dataset (adult males and females). Loadings for the last two PCs not included.

| Variable | PC1 | PC2 | PC3 | PC4 | PC5 | PC6 | PC7 |
|---|---|---|---|---|---|---|---|
| Tibia length (TL) | −0.447 | 0.189 | 0.146 | −0.105 | 0.283 | −0.154 | −0.263 |
| Foot Length (FL) | −0.407 | 0.285 | −0.085 | −0.397 | 0.163 | 0.210 | −0.338 |
| Head Length (HL) | −0.281 | −0.369 | 0.414 | −0.016 | 0.598 | 0.047 | 0.295 |
| Head Width (HW) | −0.384 | −0.353 | 0.058 | −0.007 | −0.314 | −0.070 | 0.493 |
| Eye Diameter (ED) | −0.218 | −0.155 | −0.706 | 0.508 | 0.323 | 0.253 | −0.002 |
| Interorbital Distance (IOD) | −0.103 | −0.577 | −0.307 | −0.559 | −0.237 | 0.150 | −0.210 |
| Eyelid Width (EW) | −0.334 | −0.110 | 0.388 | 0.470 | −0.432 | 0.368 | −0.390 |
| Internarial Distance (IND) | −0.392 | 0.063 | −0.197 | 0.128 | −0.198 | −0.761 | −0.051 |
| Eye-Nostril Distance (E-N) | −0.290 | 0.500 | −0.123 | −0.151 | −0.224 | 0.352 | 0.536 |
| % Variance explained | 43.04 | 19.12 | 11.07 | 7.66 | 7.12 | 5.64 | 3.80 |

**Table 2 PCA loadings females only.** Loadings for the nine morphological variables from a phylogenetic PCA ran on body-size corrected residuals following a phylogenetic size-correction using GLS regression. The first seven PCs explained 96.7% of the variance, as estimated using the reduced dataset (females only). Loadings for the last two PCs not included.

| Variable | PC1 | PC2 | PC3 | PC4 | PC5 | PC6 | PC7 |
|---|---|---|---|---|---|---|---|
| Tibia length (TL) | −0.461 | 0.122 | −0.138 | −0.051 | −0.348 | −0.187 | −0.441 |
| Foot Length (FL) | −0.451 | 0.200 | 0.226 | 0.022 | −0.242 | −0.438 | −0.014 |
| Head Length (HL) | −0.209 | −0.228 | −0.694 | −0.381 | −0.177 | −0.128 | 0.446 |
| Head Width (HW) | −0.376 | −0.359 | −0.071 | 0.216 | 0.415 | 0.180 | 0.171 |
| Eye Diameter (ED) | −0.135 | −0.386 | 0.459 | −0.542 | −0.348 | 0.451 | −0.012 |
| Interorbital Distance (IOD) | −0.064 | −0.646 | 0.295 | 0.144 | 0.133 | −0.546 | 0.025 |
| Eyelid Width (EW) | −0.313 | −0.162 | −0.137 | 0.622 | −0.292 | 0.447 | −0.066 |
| Internarial Distance (IND) | −0.396 | 0.102 | −0.067 | −0.322 | 0.618 | 0.138 | −0.390 |
| Eye-Nostril Distance (E-N) | −0.350 | 0.401 | 0.349 | 0.025 | 0.101 | 0.062 | 0.649 |
| % Variance explained | 39.95 | 18.51 | 11.69 | 11.42 | 6.00 | 5.30 | 3.80 |

**Table 3 Phylogenetic generalized linear regression models.** Results from phylogenetic generalized linear regression models for SVL and other body size-corrected variables with respect to elevation. Model fitting was done with morphological data from adult females only (21 species; see Tables 1 and 2 for variable abbreviations). With the exception of SVL, other morphological variables were corrected with respect to body size prior to tests. Bold font indicates significant values.

| Model | Evol. model | λ | Coefficient | P-value | AIC | logLik |
|---|---|---|---|---|---|---|
| SVL ~ Elev. midpoint | OLS | | 0.0092 | **0.0017** | 125.04 | −61.52 |
| SVL ~ Elev. midpoint | BM | | 0.0048 | 0.1203 | 128.84 | −63.42 |
| SVL ~ Elev. midpoint | Lambda | 0.26 | 0.0082 | **0.0041** | 126.68 | −61.34 |
| TL ~ Elev. midpoint | OLS | | <−0.0001 | **0.0116** | −73.2 | 37.6 |
| TL ~ Elev. midpoint | BM | | <−0.0001 | **0.0340** | −76.46 | 39.23 |
| TL ~ Elev. midpoint | Lambda | 1.00 | <−0.0001 | **0.0340** | −74.46 | 39.23 |
| FL ~ Elev. midpoint | OLS | | <−0.0001 | **0.0186** | −78.00 | 40.00 |
| FL ~ Elev. midpoint | BM | | <−0.0001 | 0.0544 | −74.82 | 38.41 |
| FL ~ Elev. midpoint | Lambda | 0.64 | <−0.0001 | **0.0253** | −75.94 | 39.97 |
| HL ~ Elev. midpoint | OLS | | <−0.0001 | **0.0316** | −86.36 | 44.18 |
| HL ~ Elev. midpoint | BM | | <−0.0001 | 0.2207 | −83.82 | 42.91 |
| HL ~ Elev. midpoint | Lambda | 0.00 | <−0.0001 | **0.0316** | −84.36 | 44.18 |
| E-N ~ Elev. midpoint | OLS | | <−0.0001 | **0.0007** | −144.18 | 73.09 |
| E-N ~ Elev. midpoint | BM | | <−0.0001 | **0.0014** | −138.00 | 70.00 |
| E-N ~ Elev. midpoint | Lambda | 0.00 | <−0.0001 | **0.0007** | −142.18 | 73.09 |
| IND ~ Elev. midpoint | OLS | | <−0.0001 | **<0.0001** | −146.70 | 74.35 |
| IND ~ Elev. midpoint | BM | | <−0.0001 | **<0.0001** | −142.56 | 72.28 |
| IND ~ Elev. midpoint | Lambda | 0.00 | <−0.0001 | **<0.0001** | −144.70 | 74.35 |

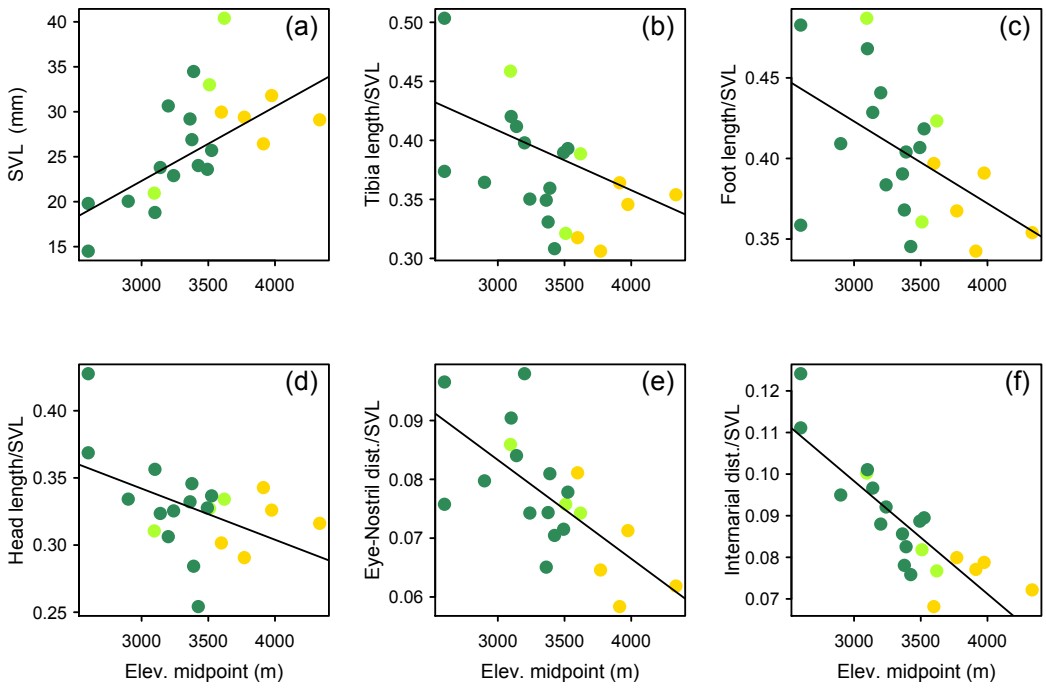

**Figure 7  Multipanel plot morphology vs elevation.** Body size and body shape in *Phrynopus* vary across elevations. Body size (A) tends to increase with increasing elevation. In contrast, when controlling for body size, species at higher elevations tend to be characterized by shorter limbs (B, C), shorter head (D), shorter snout (E), and narrower snout tips (F) than species living at lower elevations. Species are color-coded according to habitat use: Andean grassland (yellow), montane forest (dark green), and both habitats (light green). The regression lines reflect the phylogenetic correction. Based on dataset containing measurements from adult females only.

(10°22.772′S, 75°27.717′W; 3,000 m a.s.l.), Yanachaga-Chemillén National Park (Sector San Daniel), Distrito de Huancabamba, Provincia de Oxapampa, Departamento de Pasco, Peru, 2012)—*Padial, Grant & Frost, 2014*, p. 124; *Chávez et al., 2015*, p. 18, 21, 22; *De la Riva et al., 2017*, p. 31; *Lehr & Rodriguez, 2017*, p. 333, 338; *Rodríguez & Catenazzi, 2017*, p. 383, 390, 399, 402, 403, Table 2.

*Duellman & Hedges (2008)* described *Phrynopus tribulosus* based on a single male collected in 1987, *Chaparro, Padial & De la Riva (2008)* described *P. nicoleae* based on a single female collected in 2007, and *Lehr, Moravec & Cusi (2012)* described *P. curator* based on a single female collected in 2012. *Phrynopus tribulosus* is known from three localities and an elevation from 2,600 to 3,589 m. Even though *Chaparro, Padial & De la Riva (2008)* and *Lehr, Moravec & Cusi (2012)* compared in their diagnoses the new species with *P. tribulosus*, sufficient morphological differences were noted to justify the descriptions. Some of the characters mentioned for *P. tribulosus* by *Duellman & Hedges (2008)* such as skin on venter smooth (areolate according to *Chaparro, Padial & De la Riva, 2008* and *Lehr, Moravec & Cusi, 2012*) or dorsolateral folds absent (elongate dorsolateral warts forming a long discontinuous row that does not fuse to form a fold according to *Chaparro, Padial & De la Riva, 2008*, ridges forming discontinuous dorsolateral fold according to *Lehr,*

*Moravec & Cusi, 2012*) are likely the result of long storage of the specimen in preservative. Whereas other differences such as absence of dentigerous processes of vomers as noted by *Duellman & Hedges (2008)* and *Lehr, Moravec & Cusi (2012)* but noted as present by *Chaparro, Padial & De la Riva (2008)* likely reflect intraspecific variation. Singleton species descriptions can be problematic when additional discoveries of specimens are difficult to assign because variation for the species is not covered or the species was not accurately described (see *Lim, Balke & Meier, 2012*; *Köhler & Padial, 2016*; *Lehr & Rodriguez, 2017*). Therefore, we have updated the species diagnosis of *P. tribulosus* based on *Duellman & Hedges (2008)*, *Chaparro, Padial & De la Riva (2008)*, and *Lehr, Moravec & Cusi (2012)* as follows.

**Diagnosis**

A species of *Phrynopus* having the following combination of characters: (1) Skin on dorsum shagreen with small scattered tubercles, middorsum with prominent ridges, often Y- or X-shaped, flanks shagreen with small scattered tubercles, skin on venter weakly areolate; ridges forming discontinuous dorsolateral fold; discoidal fold absent, weak thoracic fold present; (2) tympanic membrane and tympanic annulus absent; (3) snout rounded in dorsal and lateral views; (4) upper eyelid with or without enlarged tubercles; width of upper eyelid narrower than IOD; cranial crests absent; (5) dentigerous processes of vomers present or absent; (6) vocal slits and nuptial pads absent; (7) Finger I shorter than or equal to Finger II; tips of digits rounded; (8) fingers without lateral fringes; (9) ulnar tubercles absent; (10) heel with a distinct conical tubercle; inner tarsal fold absent; outer edge of tarsus with row of subconical tubercles; (11) outer metatarsal tubercle rounded, larger or equal in size of ovoid inner metatarsal tubercle; supernumerary plantar tubercles weakly defined; (12) toes without lateral fringes; basal webbing absent; Toe V slightly shorter or equal to Toe III; toe tips rounded, about as large as those on fingers; (13) in life, dorsum green with brown markings, tan with black irregular stripes and bluish-gray tubercles or reddish brown with dark gray and yellowish-brown mottling; venter gray, marmorated with small, brown, tan and metallic blue blotches or venter gray with pale gray mottling and brownish-cream flecks around posterior half of belly; groin with dark brown blotch, tan with abundant bluish-white spots and an orange spot or groin brown and gray mottled; (14) SVL in single male 15.2 mm, in females 20.7–21.2 ($n = 2$) mm.

## DISCUSSION

We inferred a molecular phylogeny of *Phrynopus* and found that two nominal species, *Phrynopus curator* and *Phrynopus nicoleae* are junior synonyms of *Phrynopus tribulosus*. All three nominal taxa are found in the montane forest and Andean grassland of Pasco region, in sites located 30–50 km from each other. The elevational range of *P. tribulosus* was updated as a result of this taxonomic change. Therefore, the currently known elevational distribution of *P. tribulosus* ranges from 2,600 to 3,589 m. The high genetic similarity between *P. nicoleae* and *P. tribulosus* was originally identified by *De la Riva et al. (2017)*, but no formal taxonomic action was proposed. Considering the new evidence, we synonymized both *P. curator* and *P. nicoleae* under *P. tribulosus*.

We identified a single evolutionary transition that involved the loss of both the tympanic membrane and tympanic annulus (Fig. 4). Within *Phrynopus*, only three species exhibit both a tympanic membrane and tympanic annulus. These species, *P. auriculatus*, *P. peruanus*, and an undescribed species that is sister to *P. peruanus*, represent lineages that diverged from the rest early on in the evolutionary history of the group. Previous studies (*Lehr, 2007a*; *Lehr, 2007b*; *Duellman & Lehr, 2009*) had suggested that *P. montium* exhibits a tympanic annulus (visible beneath the skin), although this was not specified in the original description of this species (Shreve, 1938). Additionally, inspection of recently collected material from the type locality showed that no tympanic annulus and no tympanic membrane are present in this species (*von May, 2017*). Inspection of X-ray µCT images of the skull of *P. montium* showed that the columella has been completely lost in *Phrynopus montium*. This result supports the general prediction of *Pereyra et al. (2016)* that the absence of columella implies the absence of both the tympanic membrane and tympanic annulus, and also corroborates a recent study confirming the absence of tympanic annulus in *P. montium* (*von May, 2017*).

The absence of tympanic middle ear structures in most species of *Phrynopus* is correlated with the absence of advertisement calls or other vocalizations in members of this clade. To date, the only species known to produce advertisement calls are *P. auriculatus*, *P. peruanus*, and *Phrynopus* spI (*Duellman & Lehr, 2009*; E Lehr & R von May, pers. obs., 2012–2014). Thus, given that most species of *Phrynopus* do not produce calls and lack the tympanic membrane and annulus, their communication mechanism remains unknown. Taking into account the ideas proposed by *Endler (1993)*, the loss of the tympanic middle ear (sensory system) and advertisement calls (communication signal) in >90% of species of *Phrynopus* may have resulted in the development of a new mode of communication. It is possible that earless species of *Phrynopus* may have evolved a very different signal structure (e.g., chemical signals) and sensory system (e.g., olfactory) from that observed in most other anuran taxa. Some anurans rely on either chemical or visual signals for conspecific communication, and there is evidence that some groups developed modified olfactory structures to detect chemical signals in the environment (*Belanger & Corkum, 2009*). Aquatic pheromone communication in anurans has been documented in several species of at least four families (Ascaphidae, *Asay, Harowicz & SU, 2005*; Hylidae, *Wabnitz et al., 1999*; Leptodactylidae, *King et al., 2005*; Pipidae, *Rabb & Rabb, 1963*, *Pearl et al., 2000*). Knowledge about chemosensory communication in anurans that are primarily terrestrial is more limited. A potentially valuable research direction may be to examine the development of the vomeronasal organ in terrestrial breeding frogs, which is linked to the olfactory system (*Jermakowicz et al., 2004*).

We identified several species pairs where one species inhabits the Andean grassland and the other montane forest, suggesting that habitat shifts may be correlated with shifts in elevational distribution. As suggested by studies focusing on ecological gradients (e.g., *Schluter, 2001*), these shifts may reflect differences in selection pressures between montane forest and Andean grassland. However, additional data (i.e., more species pairs) are needed to formally test this hypothesis.

We were interested in examining the relationship between body size and elevation in terrestrial breeding frogs because it had been suggested that larger body size in terrestrial breeding frogs might be common in high-elevation habitats (*Hedges, 1999*; *Gonzalez-Voyer et al., 2011*), yet the evidence was very limited. We also wanted to examine the relationship because the large majority of amphibians analyzed to date do not follow Bergmann's rule (*Adams & Church, 2007*). However, previous studies showing no thermal body size cline in anurans (*Ashton, 2002*; *Ashton, 2004*; *Schäuble, 2004*; *Adams & Church, 2007*) primarily focused on temperate taxa and did not include any strabomantids, a group containing over 700 species. Our finding of a body size cline along elevation in *Phrynopus* is consistent with Bergmann's rule, which predicts that organisms tend to be larger in colder climates (*Bergmann, 1847*; *Mayr, 1956*), and corroborates previous findings showing this pattern in some species of terrestrial breeding frogs (*Hedges, 1999*; *Gonzalez-Voyer et al., 2011*) and other amphibian taxa (*Ashton, 2002*; *Schäuble, 2004*). It is also worth noticing that not all strabomantid frogs may follow the observed trend of increasing body size with increasing elevation, because some high-elevation species (e.g., *Noblella pygmaea*) are among the smallest terrestrial vertebrates in the Andes (*Lehr & Catenazzi, 2009*).

Our findings suggest that ecomorphology in *Phrynopus* varies across elevation. Our analyses of a phylogenetic PCA (phylomorphospace) support the hypothesis that species living in Andean grassland exhibit a phenotype that is different than that of species living in montane forest or species found in both habitats. Our tests using phylogenetic generalized linear regression models suggest that species living at higher elevations tend to have larger body size than those distributed at lower elevations. Additionally, species living at higher elevations tend to develop shorter limbs, shorter head, and shorter snout than those living at lower elevations. All tested relationships were significant both when controlling for phylogeny under the lambda transform model and also under pure OLS regression, and OLS was the best-supported model in five cases. Yet some relationships were not significant under the Brownian motion model (Table 3). It has been suggested that the proportional reduction of limb length in some species of terrestrial breeding frogs is related, at least in part, to the primary mode of locomotion (*Duellman & Lehr, 2009*). Our own observations of locomotion of field-captured species of *Phrynopus*, as well as several other high-elevation strabomantids (e.g., *Bryophryne*, *Noblella*), indicate that they primarily walk, as opposed to hop or jump (E Lehr & R von May, pers. obs., 2012–2014; *Duellman & Lehr, 2009*); and only one species (*P. horstpauli*) is arboreal (*Lehr, Köhler & Ponce, 2000*). However, given that most species of *Phrynopus* move primarily by walking, and considering that most of species use similar microhabitats along different elevations (i.e., moss and root mats in montane forest; moss and bunchgrass in Andean grassland) it is unlikely that locomotion is the main factor underlying the observed cline in body shape at different elevations.

## CONCLUSIONS

This study develops our understanding about the links between ecological divergence and morphological diversity of a group of Neotropical frogs that belong to the most diverse amphibian family in the Tropical Andes. Although most Strabomantidae have tympanic

middle ear, members of different genera living at high elevations have experienced the loss of tympanic membrane and tympanic annulus (*Duellman & Lehr, 2009*). Given that the group contains 10% of the world's amphibian species, further studies including additional eared and earless taxa will reveal how many independent losses of the tympanic middle ear have occurred in this large radiation. Additionally, we documented a significant pattern of increasing body size and changes in body shape with increasing elevation. The body size cline has been observed in several terrestrial breeding frogs and other amphibian taxa, although some authors have suggested that most amphibians do not follow Bergmann's rule (*Ashton, 2002*; *Ashton, 2004*; *Schäuble, 2004*; *Adams & Church, 2007*). We believe that one reason for this discrepancy is that tropical amphibians were largely underrepresented in previous studies, which primarily focused on temperate taxa. Terrestrial breeding frogs are amenable for testing hypotheses such as Bergmann's rule, because of several reasons, including high species diversity and broad elevational distribution. Given that previous tests have included amphibian taxa distributed over narrower elevational gradients, further studies should include additional clades of terrestrial breeding frogs to examine Bergmann's rule predictions. Integrating trait data such as the loss of tympanic middle ear and the loss of advertisement calls with eco-morphological data will shed light on the potential causes and ecological consequences of earlessness in Neotropical terrestrial breeding frogs. Studies that link habitat use, topography, phylogenetic history, and trait divergence will be key for explaining the elevational gradient of species diversity in tropical mountains.

## ACKNOWLEDGEMENTS

Jesús H. Córdova kindly provided a loan of specimens from the Museo de Historia Natural, Universidad Nacional Mayor de San Marcos, Lima, Peru, and Juan Carlos Cusi provided additional support at the museum. We thank Lydia Smith and the Evolutionary Genomics Laboratory at the Museum of Vertebrate Zoology (UC Berkeley) for facilitating molecular laboratory work. We thank Michelle Lynch and Erin Westeen for their help in generating μCT images included in this paper. We thank J. Mitchell for providing some of the scripts used in the analyses, and the members of the Rabosky Lab for providing helpful comments and suggestions. We also thank two anonymous reviewers for providing constructive and helpful comments on the manuscript.

### Funding

Research was supported with grants from the National Science Foundation (Postdoctoral Research Fellowship DBI-1103087), the American Philosophical Society, and the National Geographic Society Committee for Research and Exploration (Grant # 9191-12) to Rudolf von May, and by a fellowship from the David and Lucile Packard Foundation (Daniel L. Rabosky). Edgar Lehr was funded by National Geographic Society Science and Exploration Europe (GEFNE13-11) and Illinois Wesleyan University provided a Junior Faculty Leave in 2012. The funders had no role in study design, data collection and analysis, decision to publish, or preparation of the manuscript.

## Grant Disclosures

The following grant information was disclosed by the authors:

National Science Foundation Postdoctoral Research Fellowship: DBI-1103087.

American Philosophical Society.

National Geographic Society Committee for Research and Exploration: # 9191-12.

David and Lucile Packard Foundation.

National Geographic Society Science and Exploration Europe: GEFNE13-11.

## Competing Interests

The authors declare there are no competing interests.

## Author Contributions

- Rudolf von May conceived and designed the experiments, performed the experiments, analyzed the data, contributed reagents/materials/analysis tools, prepared figures and/or tables, authored or reviewed drafts of the paper.
- Edgar Lehr performed the experiments, contributed reagents/materials/analysis tools, authored or reviewed drafts of the paper.
- Daniel L. Rabosky analyzed the data, contributed reagents/materials/analysis tools, authored or reviewed drafts of the paper.

## Animal Ethics

The following information was supplied relating to ethical approvals (i.e., approving body and any reference numbers):

Use of vertebrate animals was approved by the Animal Care and Use Committee of the University of California (ACUC #R278-0412, R278-0413, and R278-0314).

## Field Study Permissions

The following information was supplied relating to field study approvals (i.e., approving body and any reference numbers):

Research and collecting permits were issued by the Dirección General Forestal y de Fauna Silvestre (DGFFS) and the Servicio Nacional Forestal y de Fauna Silvestre (120-2012-AG-DGFFS-DGEFFS, 064- 2013-AG-DGFFS-DGEFFS, 292-2014-AG-DGFFS-DGEFFS, R.D.G. No 029-2016-SERFOR-DGGSPFFS, R.D.G. 405-2016-SERFOR-DGGSPFFS, and Contrato de Acceso Marco a Recursos Genéticos, No 359-2013-MINAGRI-DGFFS-DGEFFS) and the Servicio Nacional de Areas Naturales Protegidas (No 001-2012-SERNANP-JEF). Export permits were issued by the Ministerio del Ambiente, Lima, Peru.

## DNA Deposition

The following information was supplied regarding the deposition of DNA sequences:

GenBank. GenBank accession numbers can be found in Table S2.

## Data Availability

Phylogenetic trees can be found in the Supplemental Information.

## Supplemental Information

Supplemental information for this article can be found online at http://dx.doi.org/10.7717/peerj.4313#supplemental-information.

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

## FURTHER READING

**Bossuyt F, Milinkovitch MC. 2000.** Convergent adaptive radiations in Madagascan and Asian ranid frogs reveal covariation between larval and adult traits. *Proceedings of the National Academy of Sciences of the United States of America* **97**:6585–6590 DOI 10.1073/pnas.97.12.6585.

**Lehr E, Aguilar C, Córdova JH. 2002.** Morphological and ecological remarks on *Phynopus kauneorum* (Amphibia, Anura, Leptodactylidae). *Zoologische Abhandlungen Museum Für Tierkunde Dresden* **52**:71–75.

**Lehr E, Aguilar C, Köhler G. 2002.** Two sympatric new species of *Phrynopus* (Anura: Leptodactylidae) from a cloud forest in the Peruvian Andes. *Journal of Herpetology* **36**:208–216 DOI 10.1670/0022-1511(2002)036[0208:TSNSOP]2.0.CO;2.

**Meyer CP, Geller JB, Paulay G. 2005.** Fine scale endemism on coral reefs: archipelagic differentiation in turbinid gastropods. *Evolution* **59**:113–125 DOI 10.1111/j.0014-3820.2005.tb00899.x.

**Navas CA. 2005.** Patterns of distribution of anurans in high Andean tropical elevations: insights from integrating biogeography and evolutionary physiology. *Integrative Comparative Biology* **46**:82–91.

**Palumbi SR, Martin A, Romano S, McMillan WO, Stice L, Grabowski G. 1991.** The simple fool's guide to PCR. version 2.0. Honolulu: Dept. Zoology, Univ. Hawaii. Privately published document compiled by S Palumbi.

**Paradis E, Claude J, Strimmer K. 2004.** APE: analyses of phylogenetics and evolution in R language. *Bioinformatics* **20**:289–290 DOI 10.1093/bioinformatics/btg412.