# Peer review of "Evolutionary radiation of earless frogs in the Andes: molecular phylogenetics and habitat shifts in high-elevation terrestrial breeding frogs"

_PeerJ, doi:10.7717/peerj.4313_

## Round 0.1 · original submission · Minor Revisions

Both reviewers were enthusiastic about your paper, but also raised a number of issues that need to be addressed before the paper can be accepted, particularly some specific methodological issues raised by Reviewer 1. Moreover, please notice the commented pdf by Reviewer 2, including the need to include taxonomic changes in the body of the manuscript and not as an Appendix.

Reviewer 1 ·

Basic reporting

no comment

Experimental design

The paper needs to be more clearly written throughout to explicitly state the questions and hypotheses tested, the rationale behind testing these hypotheses in this group, the conclusions that can be drawn from these results (within this specific group), and the interest to a broader readership.

Validity of the findings

In general, the manuscript adds to the general knowledge of species’ relationships and how habitat and morphology can shift among closely related species.
I enjoyed the presentation of data but had some trouble deciphering the overarching questions and hypotheses.
The rationale and benefit to the literature need to be more clearly stated in both the introduction and discussion.

Additional comments

Specific notes below:

Throughout the paper the authors sometimes put “eared” and “earless” in quotations and other times do not. Please watch this and be consistent.

The term ‘earless’ has been used to describe complete loss of all middle ear structures by some authors and is also commonly used to indicate only a lack of tympanic membrane. Please define earless for your paper and be consistent in using it only when appropriate for your definition.

Throughout the paper watch consistency of putting et al. in italics or not.

Introduction:
Line 46: Please specify that “the tympanic membrane aids in transmission of” airborne vibrations. A separate system (not including the tympanic membrane) is used for substrate borne vibrations.

End of first paragraph: The opening paragraph ends with an unanswered question regarding middle ear loss, however, this question is not addressed with the data in this paper. It would be better to specifically introduce the questions and hypotheses addressed by these data (including morphological evolution more broadly and habitat shifts).

Line 75-77: There is a lot of evidence to suggest that some alternative hearing pathways are ancestral to anurans and have not evolved in response or association with loss of the tympanic middle ear. Regardless, we cannot currently be sure that extratympanic pathways have evolved in concert with tympanic middle ear loss. All we know is that low frequency hearing is retained in most earless species and in a few earless species there are clearly alternative hearing mechanisms that allow high frequency hearing.

Line 77: Please rephrase “evolved… to” because structures are not evolved “to” do anything.

Line77-78: references appear out of order

Paragraph that starts on Line 98: The body size and other morphological data come out of nowhere and the questions associated with these hypotheses should be presented towards the forefront of the paper instead of only focusing on middle ear loss at the top of the manuscript.
Do these morphological analyses tie into your middle ear loss predictions?

Methods:
246: The ancestral tympanum condition is not determined by stochastic character mapping. The result of this analysis is simply a hypothesis of tympanum evolution for the group.

Line 251: “we used phylogenetic principle components analysis...” Again, body size and other morphology analyses come out of nowhere and the term “morphological data” is vague.

Line 253: The morphological traits should be corrected for body size in a way that controls for phylogeny (see Revell 2009 - SIZE-CORRECTION AND PRINCIPAL COMPONENTS FOR INTERSPECIFIC COMPARATIVE STUDIES).

Results:
Line 305: for the PCA analyses you should quantify shape differences (or lack of shape differences) between eared and earless species and among habitats.
You could do this by comparing morphology among groups (different averages or medians) and/or comparing evolutionary rates of morphology (different amounts of shape change).
Also – you did not set up in the introduction why you would expect different morphologies among eared and earless species. This test should be justified or removed.

Line 321: which of these relationships are significant when controlling for phylogenetic relationships and which are no longer significant? Given some of your relationships do not hold when controlling for phylogeny it is best to be clear what relationships you are confident in (always show a relationship or show a relationship in the most supported model (lowest AIC)) and which relationships are less reliable.

Discussion:
Line 368: This paragraph does not clearly explain what the relationship between habitat, elevation distribution, and body size mean. The paragraph itself jumps around and ends on a note that leaves the reader wondering if the relationships among these craugastorids matters.

Line 386-387: Again, when discussing which morphological associations with elevation it would be best to only point out the relationships that are supported by either all models or at least the best supported model.

Line 412: why would including terrestrial breeding frogs matter? Would you predict terrestrial breeding frogs would follow Bergmann’s rule more so than other frogs? Why?

Supplemental materials:
Some of your museum prefixes in the museum numbers column are not defined in the legend.

Reviewer 2 ·

Basic reporting

In accordance.

Experimental design

In accordance.

Validity of the findings

In accordance.

Additional comments

Here I return my review of the manuscript entitled “Evolutionary radiation of earless frogs in the Andes: molecular phylogenetics and habitat shifts in high-elevation terrestrial breeding frogs” by Rudolf vom May, Edgar Lehr, and Daniel L. Rabosky. The authors present a new phylogeny of the genus Phrynopus with 11 species never tested before in a phylogenetic context, including the type species, P. peruanus. The authors find that Phrynopus is monophyletic and includes a clade of “eared” and a clade of “earless” frogs, and based on their topology propose P. curator Lehr, Moravec & Cusi, 2012 and P. nicoleae Chaparro, Padial & de la Riva as junior synonyms of P. tribulosus Duellman & Hedges, 2008. Another interesting finding is that members of the genus Phrynopus increase their SVL, and decrease their forelimbs, hind limbs, and snout sizes with the increase of the elevation. The CT scan figures are amazing and the authors show the absence of tympanum membranae and annulus are connected to the loss of columella in at least one species of the genus, P. montium.
The manuscript is a great contribution, it is well-written and I suggest the acceptance considering the following changes:

(1) The authors must take off the last sentence from the abstract (Lines 32–34) because they do not talk about it anywhere else in the manuscript and also because one cannot predict what clades future analyses will reveal without any evidence.
(2) The proposed taxonomic change must be in the body of the manuscript, and not as an Appendix. The authors also must talk about these species on the introduction.
(3) The authors make some confusion in the Materials and Methods part, writing on it part of the Results section (lines 208–215, 217–218, 221–222, 236–238)
(4) The authors use the substitution rate of 16S as 0.01 bp per million years and they do not say from where they took it. It is the highest mutation rate I have ever seen for frogs and they can either explain from where they took it or take another rate from the literature (e.g. Gehara et al. 2017, phylogeography of I. parva).
(5) There are many parts in the Conclusion section which are copies of the Discussion section, the authors should take a closer look at it.
(6) Some museum abbreviations are not explained in the text, so the reader has to guess what does the acronym mean. Additionally, there are a few specimens that are not housed in any museum collection, and only the name of the collector is provided. Is this in accordance with the collection permits? Other Scientists should be able to track the studied specimens after you publish your work, as they will do with your sequences, for example.


These and other minor suggestions and misspelling checks are directly in the pdf.

Annotated reviews are not available for download in order to protect the identity of reviewers who chose to remain anonymous.

---

## Round 0.2 · accepted · Accept

I'm happy with the provided modifications. Just make sure to submit the corresponding sequences to GenBank as soon as possible to avoid delays in the publication of the paper.